# The Flip Side of the Reweighted Coin: Duality of Adaptive Dropout and Regularization

**Daniel LeJeune**
Department of Electrical
and Computer Engineering
Rice University
Houston, TX 77005
dlejeune@rice.edu

**Hamid Javadi**
Department of Electrical
and Computer Engineering
Rice University
Houston, TX 77005
hh35@rice.edu

**Richard G. Baraniuk**
Department of Electrical
and Computer Engineering
Rice University
Houston, TX 77005
richb@rice.edu

## Abstract

Among the most successful methods for sparsifying deep (neural) networks are those that adaptively mask the network weights throughout training. By examining this masking, or *dropout*, in the linear case, we uncover a *duality* between such adaptive methods and regularization through the so-called "$\eta$-trick" that casts both as iteratively reweighted optimizations. We show that any dropout strategy that adapts to the weights in a monotonic way corresponds to an effective subquadratic regularization penalty, and therefore leads to sparse solutions. We obtain the effective penalties for several popular sparsification strategies, which are remarkably similar to classical penalties commonly used in sparse optimization. Considering variational dropout as a case study, we demonstrate similar empirical behavior between the adaptive dropout method and classical methods on the task of deep network sparsification, validating our theory.

## 1 Introduction

In machine learning, it is often valuable for models to be parsimonious or *sparse* for a variety of reasons, from memory savings and computational speedups to model interpretability and generalizability. Classically, this is achieved by solving a regularized empirical risk minimization (ERM) problem of the form

$$\underset{\mathbf{w}}{\text{minimize}} \quad \mathcal{L}(\mathbf{w}) + \lambda\Omega(\mathbf{w}), \tag{1}$$

where $\mathcal{L}$ is the data-dependent empirical risk and $\Omega$ is a sparsity-inducing regularization penalty. Ideally, $\Omega(\mathbf{w})$ is equal to $\|\mathbf{w}\|_0 := \#\{i : w_i \neq 0\}$ or a function of $\|\mathbf{w}\|_0$ [6], in which case an appropriately chosen $\lambda$ balances the trade-off between the suboptimality of $\mathcal{L}$ and the sparsity of the solution. However, the use of $\|\mathbf{w}\|_0$ directly as a penalty is difficult, as it makes local search impossible, so alternative approaches are necessary for high-dimensional problems. The classical solution to this is to *relax* the problem using a smooth surrogate $\Omega$ that approximates $\|\mathbf{w}\|_0$ [5, 18].

When we know the regularization penalty $\Omega$, we can understand the types of solutions they encourage; however, many popular methods for sparsifying deep (neural) networks do not appear to fit the form (1). For example, variational dropout [27, 32], "$\ell_0$ regularization" [29], and magnitude pruning [45] are the only methods compared in a recent large-scale study by Gale et al. [15], and all achieve sparsity by adaptively *masking* or *scaling* the parameters of the network while optimizing the loss function in $\mathbf{w}$. Although these methods are well-motivated heuristically, it is unclear how to make a principled choice between them aside from treating them as black boxes and comparing their empirical performance. However, if these methods could be cast as solutions to a regularized ERM problem, then we could compare them on the basis of their regularization penalties.

35th Conference on Neural Information Processing Systems (NeurIPS 2021).

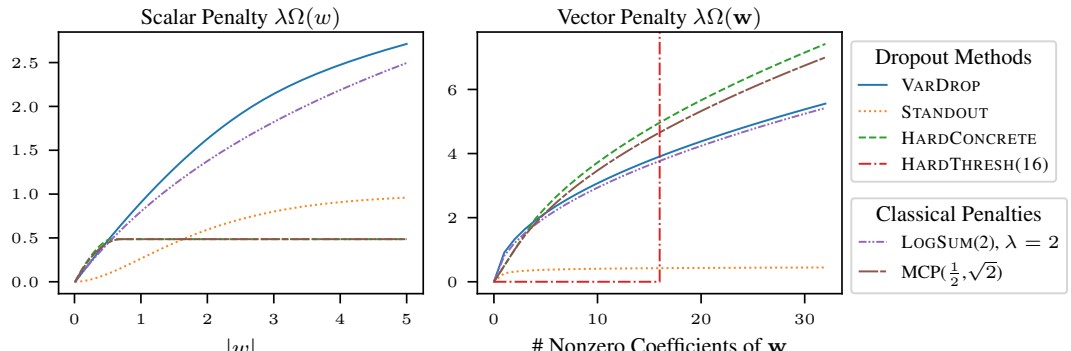

Figure 1: Sparsity-inducing behavior in regularized ERM (1) can be understood by considering the properties of the regularization penalty. Our analysis enables the computation and comparison of the effective regularization penalties of the adaptive dropout sparsity methods in Table 2. (We arbitrarily set $\lambda = 1$ unless otherwise specified.) Both VARIATIONALDROPOUT [32] and HARDCONCRETE (called "$\ell_0$ regularization") [29] bear strong resemblance to classical penalties. *Left:* Separable penalties (excluding HARDTHRESH) as a function of weight magnitude. *Right:* Penalty of a unit-norm $k$-sparse $\mathbf{w} \in \mathbb{R}^{32}$ defined by $w_j = \frac{1}{\sqrt{k}} \mathbb{1} \{j \leqslant k\}$.

We show that, in fact, these sparsity methods *do* correspond to regularization penalties $\Omega$, which we can obtain, compute, and compare. As we show in Figure 1, these penalties bear striking resemblance to classical sparsity-inducing penalties such as the LOGSUM [9] penalty and the minimax concave penalty (MCP) [44]. More broadly, we analyze *adaptive dropout* methods, which apply the dropout technique of Srivastava et al. [35] with adaptive parameters. By considering adaptive dropout in the linear setting, we uncover a *duality* between adaptive dropout methods and regularized ERM. We make this connection via the "$\eta$-trick" [4], a tool with a long history of application in sparse optimization via iteratively reweighted least squares (IRLS) (see the history presented by Daubechies et al. [11]).

We prove that all adaptive dropout methods whose amount of dropout varies monotonically with the magnitudes of the parameters induce an effective *subquadratic* (and hence sparsity-inducing [5]) penalty $\Omega$. This further supports the experimental evidence that such methods excel at inducing sparsity. We also demonstrate how to use our result to determine the effective penalty for adaptive dropout methods, using as examples the sparsity methods listed above as well as the standout method [3], which has an effective penalty on the layer *activations*, rather than the parameters, explaining why the standout method sparsifies activations. We then numerically compute the effective penalties[1] and plot them together in Figure 1.

We validate our new theory by applying variational dropout on the task of deep network sparsification, and we show that the performance is similar to non-dropout methods based on the same effective penalty. This suggests that not only is considering the effective penalty a sound means for comparing sparsity methods, but also that classical regularized ERM is itself an effective means for inducing sparsity in deep networks and serves as a valuable tool for future work on deep network sparsity.

For theoreticians, this work provides a general framework for obtaining the effective regularization penalty for adaptive dropout algorithms, enabling regularized ERM analysis tools to be applied to these methods. For practitioners, this work enables the application of classical regularization intuition when choosing between sparsifying approaches. For methods developers, this work provides a strong baseline against which new adaptive dropout methods should be compared: the effective penalty $\Omega$.

## 2 Background

**Notation.** We denote the extended reals and the non-negative reals by $\overline{\mathbb{R}} = \mathbb{R} \cup \{-\infty, \infty\}$ and $\mathbb{R}_+ = [0, \infty)$, respectively. The operator $\mathrm{diag}(\cdot)$ takes a vector and returns a matrix with that vector along the diagonal entries or takes a matrix and returns the same matrix with all non-diagonal entries set to zero. The matrix $\mathbf{J}$ denotes a matrix of all ones. We denote element-wise multiplication and exponentiation by $\odot$; i.e., $[\mathbf{u} \odot \mathbf{v}]_j = u_j v_j$ and $[\mathbf{U}^{\odot p}]_{ij} = U_{ij}^p$. Division of vectors and scalar functions of vectors denote element-wise operations. Order statistics of the magnitudes of

---

[1]Our code is available at `https://github.com/dlej/adaptive-dropout`.

elements in a vector $\mathbf{u} \in \mathbb{R}^d$ are denoted by $u_{(j)}$ such that $|u_{(1)}| \geqslant |u_{(2)}| \geqslant \ldots |u_{(d)}|$. The function $\sigma \colon \overline{\mathbb{R}} \to [0,1]$ denotes the sigmoid $\sigma(t) = 1/(1 + e^{-t})$.

## 2.1 Dropout as Tikhonov Regularization

We can understand adaptive dropout methods by breaking them down into two components: the regularizing effect of dropout itself and the effect of adaptively updating the dropout-induced regularization. For the former, it is well-known [40, 39, 35, 32] that dropout induces Tikhonov-like regularization for standard binary and Gaussian dropout. The same result holds more broadly given a second moment condition, which we need in order to be able to consider methods like HARDCON-CRETE "$\ell_0$ regularization" [29]. In general, we use the term *dropout* to refer to methods that mask or scale the weights $\mathbf{w}$ by independent "mask" variables $\mathbf{s} \sim \text{MASK}(\boldsymbol{\alpha})$ with sampling parameters $\boldsymbol{\alpha} \in [0,1]^d$. We emphasize that in this formulation, it is the *parameters* that are masked, rather than the nodes of a network as in the original formulation of dropout [35], although in linear regression there is no difference. We assume that the masks are *unbiased* (i.e., that $\mathbb{E}\left[s_j\right] = 1$), and that $\mathbb{E}\left[s_j^2\right] = \alpha_j^{-1}$. Note that with the independence assumption, this implies that $\text{Cov}(\mathbf{s}) = \text{diag}\left(\boldsymbol{\alpha}\right)^{-1} - \mathbf{I}$. Both standard unbiased binary dropout, where $s_j$ take value $\alpha_j^{-1}$ with probability $\alpha_j$ and $0$ otherwise, as well as Gaussian dropout with $s_j \sim \mathcal{N}\left(1, \alpha_j^{-1} - 1\right)$, satisfy these assumptions.

Dropout solves the optimization problem

$$\underset{\mathbf{w}}{\text{minimize}} \quad \mathbb{E}_{\mathbf{s}}\left[\mathcal{L}(\mathbf{s} \odot \mathbf{w})\right] \tag{2}$$

via stochastic optimization such as gradient descent where each iteration performs a partial optimization of $\mathcal{L}(\mathbf{s} \odot \mathbf{w})$ with a random $\mathbf{s}$. It is most insightful to consider when $\mathcal{L}$ is the loss for linear regression of target variables $\mathbf{y} \in \mathbb{R}^n$ given data $\mathbf{X} \in \mathbb{R}^{n \times d}$:

$$\mathcal{L}(\mathbf{w}) = \frac{1}{2n}\|\mathbf{y} - \mathbf{X}\mathbf{w}\|_2^2. \tag{3}$$

In this case, under our assumptions on $\mathbf{s}$, we recover the known Tikhonov result [e.g., 32]

$$\mathbb{E}_{\mathbf{s}}\left[\mathcal{L}(\mathbf{s} \odot \mathbf{w})\right] = \mathcal{L}(\mathbf{w}) + \frac{1}{2}\mathbf{w}^{\top}\left(\left(\text{diag}\left(\boldsymbol{\alpha}\right)^{-1} - \mathbf{I}\right) \odot \frac{1}{n}\mathbf{X}^{\top}\mathbf{X}\right)\mathbf{w}. \tag{4}$$

For simplicity, we assume that the data is standardized, or that $\text{diag}\left(\frac{1}{n}\mathbf{X}^{\top}\mathbf{X}\right) = \mathbf{I}$, which is inexpensive to satisfy in practice, but our analysis can be extended to the general case. Under this assumption, we see that dropout elicits a diagonal Tikhonov regularization with scale $\text{diag}\left(\boldsymbol{\alpha}\right)^{-1} - \mathbf{I}$.

## 2.2 The "$\eta$-trick"

The second component of adaptive dropout is the adaptive update of the dropout-induced regularization. To understand this, we introduce the so-called "$\eta$-trick" [4] applied to the regularization penalty in (1). By introducing an auxiliary variable $\boldsymbol{\eta} \in \mathcal{H} \subseteq \overline{\mathbb{R}}_+^d$, we can replace $\Omega(\mathbf{w})$ with a dual formulation as a function of $\boldsymbol{\eta}$ that majorizes $\Omega(\mathbf{w})$ and is quadratic in $\mathbf{w}$ for fixed $\boldsymbol{\eta}$. In other words, we can find a function $f : \mathcal{H} \subseteq \overline{\mathbb{R}}_+^d \to \overline{\mathbb{R}}$ such that

$$\Omega(\mathbf{w}) = \min_{\boldsymbol{\eta} \in \mathcal{H}} \frac{1}{2}\left(\mathbf{w}^{\top}\text{diag}\left(\boldsymbol{\eta}\right)^{-1}\mathbf{w} + f(\boldsymbol{\eta})\right). \tag{5}$$

If such a function $f$ exists, then the regularized ERM problem (1) can be rewritten as a joint optimization in $\mathbf{w}$ and $\boldsymbol{\eta}$ of the dual regularized ERM formulation

$$\underset{\mathbf{w},\boldsymbol{\eta}}{\text{minimize}} \quad \mathcal{L}(\mathbf{w}) + \frac{\lambda}{2}\left(\mathbf{w}^{\top}\text{diag}\left(\boldsymbol{\eta}\right)^{-1}\mathbf{w} + f(\boldsymbol{\eta})\right). \tag{6}$$

This joint optimization can be performed, for example, by alternating minimization over $\mathbf{w}$ and $\boldsymbol{\eta}$. For linear regression problems, this gives rise to the iteratively reweighted least squares (IRLS) algorithm, a popular method for sparse recovery [10, 11]. We note that iteratively reweighted $\ell_1$ schemes have also held a significant place in sparse optimization [47, 9, 42], but because dropout regularization is quadratic, we limit our consideration to iteratively reweighted $\ell_2$ (Tikhonov) regularization like (6).

Table 1: Common regularization penalties have dual formulations that are often straightforward to obtain. See Table 3 in Appendix A for a more complete table of common penalties along with the derivations.

| Penalty | $\Omega(\mathbf{w})$ | $f(\boldsymbol{\eta})$ | $\widehat{\eta}_j(\mathbf{w})$ |
|---|---|---|---|
| $\ell_1$ | $\lvert w_j \rvert$ | $\eta_j$ | $\lvert w_j \rvert$ |
| LogSum($\varepsilon$) [9] | $\log\left(\lvert w_j \rvert + \varepsilon\right)$ | $2\log\left(\frac{\sqrt{\varepsilon^2+4\eta_j}+\varepsilon}{2}\right) - \frac{\left(\sqrt{\varepsilon^2+4\eta_j}-\varepsilon\right)^2}{4\eta_j}$ | $\lvert w_j \rvert(\lvert w_j \rvert + \varepsilon)$ |
| MCP($a, \lambda$) [44] | $\begin{cases}\lvert w_j \rvert - \frac{w_j^2}{2a\lambda}, & \lvert w_j \rvert \leqslant a\lambda \\ \frac{a\lambda}{2}, & \lvert w_j \rvert > a\lambda\end{cases}$ | $\frac{a\lambda\eta_j}{\eta_j + a\lambda}$ | $\begin{cases}\frac{a\lambda\lvert w_j \rvert}{a\lambda - \lvert w_j \rvert}, & \lvert w_j \rvert < a\lambda \\ \infty, & \lvert w_j \rvert \geqslant a\lambda\end{cases}$ |
| HardThresh($k$) [7] | $\infty \mathbb{1}\left\{\lVert \mathbf{w} \rVert_0 > k\right\}$ | $0, \ \mathcal{H} = \left\{\boldsymbol{\eta} : \lVert \boldsymbol{\eta} \rVert_0 \leqslant k\right\}$ | $\infty \mathbb{1}\left\{j \in \text{Top-}k(\mathbf{w})\right\}$ |

The simplest example of a penalty $\Omega$ that fits the form (5) is $\Omega(w) = \lvert w \rvert$, for which $f(\eta) = \eta$, and it is used in the IRLS formulation of $\ell_1$ regularization. More broadly, we can consider the class of functions for which $\Omega$ is concave in $\mathbf{w}^{\odot 2}$. These functions are said to be *subquadratic*, and are known to have a sparsity-inducing effect [5]. In fact, these are the only functions $\Omega$ that can fit this form. This is because $\mathbf{w}^{\odot 2} \mapsto -2\Omega(\mathbf{w})$ is the Legendre–Fenchel (LF) transform of $-\boldsymbol{\eta}^{\odot -1} \mapsto f(\boldsymbol{\eta})$, and thus $\Omega$ is concave in $\mathbf{w}^{\odot 2}$. By the Fenchel–Moreau theorem, there is conversely a unique function $f$ satisfying (5) that is convex in $-\boldsymbol{\eta}^{\odot -1}$, which can be obtained by the LF transform. That is, $\Omega$ and $f$ are *dual* functions. Additionally, for differentiable penalties, the minimizing $\boldsymbol{\eta}$ given a fixed $\mathbf{w}$ is

$$\widehat{\boldsymbol{\eta}}(\mathbf{w}) = \left(2\nabla_{\mathbf{w}^{\odot 2}}\Omega(\mathbf{w})\right)^{\odot -1}, \tag{7}$$

which means that if $\Omega(\mathbf{w})$ has a closed-form expression, then so does the minimizer $\widehat{\boldsymbol{\eta}}(\mathbf{w})$. The simplicity of this form means that iteratively reweighted Tikhonov regularization is simple to implement for any differentiable penalty, leading to its popularity as a sparse optimization technique.

We enumerate a few penalties along with their dual formulations in Table 1. For separable penalties, we list the scalar function applied to a single element. Where $\mathcal{H}$ is restricted, we include it with $f(\boldsymbol{\eta})$.

**Hard Thresholding.** Of particular note is HardThresh, the indicator penalty of level sets of $\lVert \mathbf{w} \rVert_0$, which gives rise to the iterative hard thresholding (IHT) algorithm [7]. We remark that to the best of our knowledge, what we show in Table 1 is the first characterization of IHT as an iteratively reweighted $\ell_2$ solution to a regularized ERM problem, rather than as a heuristic algorithm. This connection is important for understanding the MagnitudePruning [45] algorithm in Section 4.4.

## 3 Adaptive Dropout as Iteratively Reweighted Stochastic Optimization

We can now consider an adaptive dropout algorithm [3, 27, 29], which both optimizes the weights $\mathbf{w}$ using dropout and updates the dropout parameters $\boldsymbol{\alpha}$ according to some update rule depending on $\mathbf{w}$. To facilitate our analysis, we can equivalently consider the algorithm to be updating parameters $\eta_j := \lambda\left(\alpha_j^{-1} - 1\right)^{-1}$, which is a monotonic and invertible function of $\alpha_j$, with

$$\alpha_j = \frac{\eta_j}{\eta_j + \lambda}. \tag{8}$$

An adaptive dropout method could resemble, for example, the following gradient descent strategy with step size parameter $\rho$ given an update function $\widehat{\boldsymbol{\eta}} : \mathbb{R}^d \to \mathcal{H}$:

$$\mathbf{w}^{t+1} = \mathbf{w}^t - \rho\nabla_{\mathbf{w}}\mathcal{L}\left(\mathbf{s}^t \odot \mathbf{w}^t\right), \quad \mathbf{s}^t \sim \text{Mask}(\boldsymbol{\alpha}^t), \quad \alpha_j^t = \frac{\widehat{\eta}_j(\mathbf{w}^t)}{\widehat{\eta}_j(\mathbf{w}^t) + \lambda}. \tag{9}$$

The stochastic update of $\mathbf{w}$ corresponds to the expected loss $\mathbb{E}_{\mathbf{s}}\left[\mathcal{L}(\mathbf{s} \odot \mathbf{w})\right]$, but we must also characterize the $\boldsymbol{\eta}$ update. We can make the assumption, which we will justify later, that $\widehat{\boldsymbol{\eta}}(\mathbf{w})$ is the minimizer given $\mathbf{w}$ of a joint objective function $\mathcal{J}(\mathbf{w}, \boldsymbol{\eta})$ with a function $f$ such that

$$\mathcal{J}(\mathbf{w}, \boldsymbol{\eta}) = \mathbb{E}_{\mathbf{s}}\left[\mathcal{L}(\mathbf{s} \odot \mathbf{w})\right] + \frac{\lambda}{2}f(\boldsymbol{\eta}). \tag{10}$$

This brings us to our main result, in which we uncover a *duality* between the adaptive dropout objective function $\mathcal{J}(\mathbf{w}, \boldsymbol{\eta})$ and an effective regularized ERM objective function as in (1).

**Theorem 1.** *If $\mathcal{L}$ has the form* (3) *for data $\mathbf{X}$ such that $\frac{1}{n}\mathrm{diag}\left(\mathbf{X}^\top\mathbf{X}\right) = \mathbf{I}$, then for every $\lambda > 0$ there exists a function $f$ such that $\widehat{\boldsymbol{\eta}}(\mathbf{w}) = \arg\min_{\boldsymbol{\eta}} \mathcal{J}(\mathbf{w}, \boldsymbol{\eta})$ if and only if there exists a function $\Omega$ that is concave in $\mathbf{w}^{\odot 2}$ such that*

$$\min_{\boldsymbol{\eta}\in\mathcal{H}} \mathcal{J}(\mathbf{w}, \boldsymbol{\eta}) = \mathcal{L}(\mathbf{w}) + \lambda\Omega(\mathbf{w}). \tag{11}$$

*Furthermore, if $f$ exists, then the corresponding $\Omega$ is unique, and if $\Omega$ exists, then there is a unique such $f$ that is convex in $-\boldsymbol{\eta}^{\odot -1}$.*

*Proof.* For linear regression under the standardized data assumption, combining the definition of $\boldsymbol{\eta}$ as a function of $\boldsymbol{\alpha}$ with (4) makes $\mathcal{J}(\mathbf{w}, \boldsymbol{\eta})$ of the form (6). Then by the properties of the LF transform[2] as discussed in Section 2.2, we obtain the existence and uniqueness of $f$ and $\Omega$. $\qquad\square$

That is, every subquadratic penalty $\Omega$ has a dual formulation via the $\eta$-trick and therefore has a corresponding adaptive dropout strategy. The converse holds as well, provided the dropout update function $\widehat{\boldsymbol{\eta}}$ can be expressed as a minimizer of the joint objective. While this may seem restrictive, if $\widehat{\boldsymbol{\eta}}$ is *separable*, we have the following general result.

**Corollary 2.** *If $\widehat{\alpha}_j(w_j) := \frac{\widehat{\eta}_j(w_j)}{\widehat{\eta}_j(w_j)+\lambda}$ is a monotonically increasing differentiable function of $|w_j|$, then there exist separable $f$ and $\Omega$ satisfying Theorem 1.*

*Proof.* Because the relationship between $\alpha$ and $\eta$ is monotonic and invertible, $\widehat{\eta}_j(w_j)$ is also monotonically increasing. Thus, using (7), $\Omega(w_j) = \int_a^{w_j^2} \frac{1}{2\widehat{\eta}(t^{1/2})}dt + C_a$, for which $\frac{\partial^2}{(\partial w_j^2)^2}\Omega(w_j) \leqslant 0$. $\quad\square$

Therefore, any adaptive dropout method that updates the dropout parameters $\boldsymbol{\alpha}$ monotonically as a function of the magnitudes of the elements of $\mathbf{w}$ is equivalent to regularized ERM with a subquadratic penalty $\Omega$. It is well-known that subquadratic penalties are sparsity-inducing [5], which agrees with the "rich get richer" intuition that larger $w_j$, which receive larger $\alpha_j$, are penalized less and therefore are more likely to grow even larger, while conversely smaller $w_j$ receive smaller $\alpha_j$ and are even more penalized. Corollary 2 holds generally for the $\eta$-trick and is not limited to adaptive dropout.

We emphasize that linear regression is the setting in which Theorem 1 gives an exact duality, but that we can still expect a similar correspondence more broadly. For instance, Wager et al. [39] showed that for generalized linear models, dropout elicits a similar effect to (4), albeit with a somewhat more complex dependence on $\mathbf{w}$ and the data. In Section 5, we empirically consider deep networks and demonstrate that the behavior is very similar between adaptive dropout and algorithms that solve the corresponding effective regularized ERM problem.

## 4  Several Sparsity Methods and Their Effective Penalties

Armed with the fact that adaptive dropout strategies correspond to regularization penalties, we now have a way to mathematically compare adaptive dropout methods. Table 2 lists the methods that we compare and a corresponding characterization of the dual formulation. These methods vary in their parameterizations, and most do not admit closed-form expressions for their effective penalties. However, given these characterizations, we can numerically compute the effective penalty and make a principled comparison, which we plot in Figure 1.

Because some of the methods we consider use slightly different variants of the dropout algorithm, we first make clear how these fit into our analysis framework.

**Reparameterization Tricks.** The problem of noise in stochastic gradients, especially when $\alpha_j$ is very small, can limit the applicability of dropout methods for sparsity. One way researchers have found to reduce variance is through "reparameterization tricks" such as the local reparameterization trick [40, 27] and the additive noise reparameterization trick [32], both used in VARIATIONALDROPOUT, which consider alternative noise models without changing the optimization problem.

To consider the MAGNITUDEPRUNING algorithm, we develop another reparameterization trick. Inspired by the additive reparameterization trick from Molchanov et al. [32], we introduce another

---

[2]Results on the LF transform do not apply to functions with discontinuities at $\{-\infty, \infty\}$, such as $-\boldsymbol{\eta}^{\odot -1} \mapsto f(\boldsymbol{\eta})$ for the $\ell_0$ and HARDTHRESH($k$) penalties, so our theorem does not prove the existence of such functions.

Table 2: Select sparsity methods and their effective penalty characterizations.

| Method | Effective Penalty Characterization |
|---|---|
| STANDOUT [3] | $\widehat{\alpha}_j\left((z_j)_+\right) = \sigma((z_j)_+)$ |
| VARIATIONALDROPOUT [32] | $f(\eta) = 2D_{\mathrm{KL}}\left(q_{\frac{\eta}{\lambda},\mathbf{w}}(\widetilde{\mathbf{w}})\|p(\widetilde{\mathbf{w}})\right) = \int_0^{\frac{\eta}{\lambda}}\frac{F(\sqrt{t/2})}{\sqrt{t/2}}\,dt$ |
| HARDCONCRETEL0NORM [29] | $\begin{cases} f(\widetilde{\boldsymbol{\eta}}) = 2\sum_j \sigma(\log(a_j) - \beta\log(-\gamma/\zeta)) \\ \widetilde{\eta}_j = \frac{\lambda\mathbb{E}[s_j]^2}{\mathbb{E}[s_j^2]-\mathbb{E}[s_j]^2}; \quad s_j \sim \mathrm{HARDCONCRETE}_{\beta,\gamma,\zeta}(a_j) \end{cases}$ |
| MAGNITUDEPRUNING($k$) [45] | $\widehat{\eta}_j(\mathbf{w}) = \infty\mathbb{1}\left\{j \in \mathrm{TOP}\text{-}k(\mathbf{w})\right\}$ |

variable $\mathbf{v}$ and add the constraint $\mathbf{v} = \mathbf{w}$, under which $\mathcal{L}(\mathbf{s}\odot\mathbf{w}) = \mathcal{L}(\mathbf{w} + (\mathbf{s}-\mathbf{1})\odot\mathbf{v})$. Then for least squares with standardized data, (4) becomes

$$\mathbb{E}_{\mathbf{s}}\left[\mathcal{L}(\mathbf{w} + (\mathbf{s}-\mathbf{1})\odot\mathbf{v})\right] = \mathcal{L}(\mathbf{w}) + \frac{1}{2}\mathbf{v}^\top\left(\mathrm{diag}\left(\boldsymbol{\alpha}\right)^{-1} - \mathbf{I}\right)\mathbf{v}. \tag{12}$$

Under this parameterization, we can now use low-variance stochastic optimization for $\mathbf{w}$ and deterministic optimization for $\mathbf{v}$, along with a strategy for enforcing the equality constraint $\mathbf{v} = \mathbf{w}$ such as ADMM [8], thus reducing the overall variance. A simple proximal variable update strategy under this reparameterization (see Appendix B) gives rise to an adaptive proximal gradient descent algorithm, which we use to describe the MAGNITUDEPRUNING algorithm.

**Biased Masks.** It is not uncommon for dropout strategies to use *biased* masks in the sense that $\mathbb{E}\left[s_j\right] \neq 1$, e.g., for $s_j \sim \mathrm{BERNOULLI}(\alpha_j)$. Under a few mild assumptions on the parameterization of this distribution, we can consider adaptive dropout in this setting as well. Suppose that $\boldsymbol{\mu} := \boldsymbol{\mu}(\boldsymbol{\alpha}) = \mathbb{E}\left[\mathbf{s}\right]$ is an invertible function of $\boldsymbol{\alpha}$. Define $\widetilde{s}_j := s_j/\mu_j$, $\widetilde{\alpha}_j := \mu_j^2/\mathbb{E}\left[s_j^2\right]$, and $\widetilde{\mathbf{w}} := \boldsymbol{\mu}\odot\mathbf{w}$. Then $\widetilde{\mathbf{s}}$ is an unbiased mask, and since $\mathbf{s}\odot\mathbf{w} = \widetilde{\mathbf{s}}\odot\widetilde{\mathbf{w}}$, the expected loss has the same form as in (4), but with $\widetilde{\boldsymbol{\alpha}}$ and $\widetilde{\mathbf{w}}$ in place of $\boldsymbol{\alpha}$ and $\mathbf{w}$. Let $\widetilde{\eta}_j := \lambda\left(\widetilde{\alpha}_j^{-1} - 1\right)^{-1}$, and suppose there exists a function $\psi$ such that $\mu_j = \psi(\widetilde{\eta}_j)$, which is true if the mapping between $\widetilde{\boldsymbol{\alpha}}$ and $\boldsymbol{\mu}$ is one-to-one. Now consider the expected dual regularized ERM formulation

$$\underset{\mathbf{w},\widetilde{\boldsymbol{\eta}}}{\mathrm{minimize}} \quad \mathcal{L}\left(\boldsymbol{\mu}\odot\mathbf{w}\right) + \frac{\lambda}{2}\left((\boldsymbol{\mu}\odot\mathbf{w})^\top\mathrm{diag}\left(\widetilde{\boldsymbol{\eta}}\right)^{-1}(\boldsymbol{\mu}\odot\mathbf{w}) + f(\widetilde{\boldsymbol{\eta}})\right),$$
$$\text{subject to} \quad \mu_j = \psi(\widetilde{\eta}_j),\ \forall j. \tag{13}$$

Thus, an adaptive dropout strategy that jointly optimizes $\widetilde{\boldsymbol{\eta}}$ and $\mathbf{w}$ given $\mathbf{s} \sim \mathrm{MASK}(\boldsymbol{\alpha})$ solves the regularized ERM problem for $\widetilde{\mathbf{w}}$, where $\boldsymbol{\alpha}$ is set according to $\widetilde{\boldsymbol{\eta}}$.

## 4.1 Standout

One of the earliest methods to use adaptive probabilities with dropout was the STANDOUT method of Ba and Frey [3], who proposed to augment a neural network with an additional parallel "standout" network of the same architecture that controlled the dropout probabilities for each activation, which they showed results in *sparse activations*. Interestingly, the authors noted that the method worked just as well if the standout net was not trained but instead simply copied the weights of the primary network at each iteration, which fits our adaptive dropout framework.

Consider a single hidden layer neural network with ReLU activations $(u)_+ = \max\{u, 0\}$. Denote the pre-activations from the first layer as $\mathbf{z} = \mathbf{W}_1\mathbf{x}$. The output of the network is given by $\widehat{y} = \mathbf{w}_2^\top\left(\mathbf{s}\odot(\mathbf{z})_+\right)$, where the mask $\mathbf{s} \sim \mathrm{MASK}\left(\sigma(\mathbf{z})\right)$ is from the standout network. Now let us consider the network as a function of the first layer activations $(\mathbf{z})_+$. Using (4) and the fact that $\sigma(t)^{-1} - 1 = e^{-t}$, we know that the expected squared loss in this setting is

$$\frac{1}{2}\mathbb{E}_{\mathbf{s}}\left[\left\|y - \mathbf{w}_2^\top\left(\mathbf{s}\odot(\mathbf{z})_+\right)\right\|_2^2\right] = \frac{1}{2}\|y - \mathbf{w}_2^\top(\mathbf{z})_+\|_2^2 + \frac{1}{2}(\mathbf{z})_+^\top\mathrm{diag}\left(e^{-\mathbf{z}}\odot\mathbf{w}_2\mathbf{w}_2^\top\right)(\mathbf{z})_+. \tag{14}$$

However, terms where $z_j < 0$ make no contribution to the effective penalty, so we can substitute $(z_j)_+$ for $z_j$ in $e^{-\mathbf{z}}$. This enables us to extract $\widehat{\boldsymbol{\eta}}((\mathbf{z})_+)$ as in Section 2.1

$$\widehat{\eta}_j\left((z_j)_+\right) = \frac{\lambda e^{(z_j)_+}}{[w_2]_j^2}. \tag{15}$$

We note that (15) represents the implicit update of $\boldsymbol{\eta}$ each iteration as the standout network weights are copied from the primary network. By Corollary 2, we know that STANDOUT is therefore an adaptive dropout strategy with a subquadratic effective penalty $\Omega$ applied to the activations $(\mathbf{z})_+$:

$$\Omega((\mathbf{z})_+) = \frac{1}{2\lambda} \sum_j \int_0^{(z_j)_+^2} [w_2]_j^2 e^{-\sqrt{t}} dt = \frac{1}{\lambda} \sum_j [w_2]_j^2 \left(1 - ((z_j)_+ + 1)\, e^{-(z_j)_+}\right). \quad (16)$$

This penalty is a smooth approximation to the $\ell_0$ penalty, as shown in Figure 1, and thus naturally sparsifies the activations as $\mathbf{W}_1$ is trained. Additionally, $\lambda\Omega((\mathbf{z})_+)$ has no dependence on $\lambda$, just as the adaptive dropout parameter update has no dependence on $\lambda$.

### 4.2 Variational Dropout

Variational Bayes interpretations of dropout arose with the works of Kingma et al. [27] and Gal and Ghahramani [14], providing an automated means of choosing the dropout parameter through variational inference. Molchanov et al. [32] later showed that if each weight is allowed to have its own dropout parameter, VARIATIONALDROPOUT results in sparse solutions.

The authors of VARIATIONALDROPOUT consider maximum *a posteriori* inference of $\mathbf{w}$ using the improper log-uniform prior, for which $p(\log|w_j|) \propto c$ or $p(|w_j|) \propto 1/|w_j|$. This is equivalent to solving the LOGSUM(0)-regularized ERM problem

$$\underset{\mathbf{w}}{\text{minimize}} \quad \mathcal{L}(\mathbf{w}) + \sum_j \log(|w_j|). \quad (17)$$

Obviously, this is minimized by taking $\mathbf{w} = \mathbf{0}$ for common loss functions, which is uninteresting, and Hron et al. [20] discuss other issues with the framework. However, VARIATIONALDROPOUT does not solve the problem in (17), and instead performs inference over a new variable $\widetilde{\mathbf{w}}$, using an approximate posterior[3] $q_{n\boldsymbol{\eta},\mathbf{w}}(\widetilde{\mathbf{w}})$ such that $\widetilde{\mathbf{w}} = \mathbf{s} \odot \mathbf{w}$, where $\mathbf{s} \sim \text{MASK}(\boldsymbol{\alpha})$ for Gaussian dropout, having $\boldsymbol{\alpha}$ and $\boldsymbol{\eta}$ related by (8). Then maximizing the variational lower bound is equivalent to solving

$$\underset{\mathbf{w},\boldsymbol{\eta}}{\text{minimize}} \quad \mathbb{E}_{\mathbf{w}}\left[\mathcal{L}(\mathbf{s} \odot \mathbf{w})\right] + \frac{1}{n} D_{\text{KL}}\left(q_{n\boldsymbol{\eta},\mathbf{w}}(\widetilde{\mathbf{w}}) \| p(\widetilde{\mathbf{w}})\right), \quad (18)$$

where $D_{\text{KL}}(\cdot\|\cdot)$ is the Kullback–Leibler (KL) divergence. With the choice of the log-uniform prior, the KL divergence has no dependence on $\mathbf{w}$ and is purely a function of $\boldsymbol{\eta}$ [27]. Hron et al. [20] show that this quantity can be expressed in terms of Dawson's integral $F(u) = e^{-u^2} \int_0^u e^{t^2} dt$:

$$D_{\text{KL}}\left(q_{n\eta,w}(\widetilde{w}) \| p(\widetilde{w})\right) = \int_0^{n\eta} \frac{F(\sqrt{t/2})}{\sqrt{2t}} dt, \quad (19)$$

which we can compute via numerical integration using standard software implementations of $F(u)$. Clearly then, the Monte Carlo variational Bayes optimization of $\mathbf{w}$ and $\boldsymbol{\eta}$ in VARIATIONALDROPOUT is an adaptive dropout strategy with $\lambda = \frac{1}{n}$ and $f(\boldsymbol{\eta}) = 2D_{\text{KL}}\left(q_{\frac{1}{\lambda}\boldsymbol{\eta},\mathbf{w}}(\widetilde{\mathbf{w}}) \| p(\widetilde{\mathbf{w}})\right)$, and we can numerically compute the effective penalty $\Omega$. VARIATIONALDROPOUT also uses additive and local reparameterization tricks to reduce variance.

As shown in Figure 1, the effective penalty for $\lambda = 1$ is quite similar to the LOGSUM(2) penalty for $\lambda = 2$. We explore this connection further experimentally in Section 5.

### 4.3 Hard Concrete "$L_0$ Regularization"

While the desired regularization is typically an $\ell_0$ penalty, its non-differentiability makes it difficult to perform gradient-based optimization, which motivates finding smooth approximations to the $\ell_0$ penalty. Louizos et al. [29] propose to first consider $\mathcal{L}(\bar{\mathbf{w}}) + \frac{\lambda}{2}\|\bar{\mathbf{w}}\|_0$, using the decomposition $\bar{\mathbf{w}} = \mathbf{s} \odot \mathbf{w}$ for binary $\mathbf{s}$, where $\|\bar{\mathbf{w}}\|_0 = \|\mathbf{s}\|_0$.

We take this opportunity to note that we can consider this problem using adaptive dropout with $f(\boldsymbol{\eta}) = \mathbb{E}_{\mathbf{s}}\left[\|\mathbf{s}\|_0\right] = \sum_j \alpha_j$ if we use biased masks $s_j \sim \text{BERNOULLI}(\alpha_j)$. Interestingly, when $\lambda = 1$, the resulting penalty $\Omega$ turns out to be equal to the minimax concave penalty [44] MCP$(1,1)$, giving the MCP an additional interpretation as a stochastic binary smoothing of the $\ell_0$ penalty.

---

[3]The parameter $\alpha$ from Molchanov et al. [32] is distinct from ours but is related to $\eta$ by $\alpha^{-1} = n\eta$.

However, Louizos et al. [29] do not solve this problem, but instead choose a distribution for $\mathbf{s}$ such that the random variables $\mathbf{s}$ themselves are differentiable with respect to $\boldsymbol{\alpha}$, allowing them to extract gradient information for $\boldsymbol{\alpha}$ from $\mathcal{L}(\mathbf{s} \odot \mathbf{w})$. They propose the following biased distribution, called the $\text{HARDCONCRETE}_{\beta, \gamma, \zeta}(a_j)$ distribution:

$$z_j = \sigma\left((\log u_j - \log(1 - u_j) + \log a_j)/\beta\right), \quad u_j \sim \text{UNIFORM}[0, 1] \tag{20}$$

$$s_j = \min\left\{1, \max\left\{0, (\zeta - \gamma)z_j + \gamma\right\}\right\}. \tag{21}$$

They then use as their penalty $f(\widetilde{\boldsymbol{\eta}}) = 2\sum_j \Pr(s_j > 0) = 2\sigma(\log(a_j) - \beta\log(-\gamma/\zeta))$ under the biased dropout reparameterization. The relationships between $\mathbb{E}[s_j]$, $a_j$, $\widetilde{\alpha}_j$, and $\widetilde{\eta}_j$ are all invertible, so we can numerically determine the corresponding penalty $\Omega$.

We plot the resulting penalty in Figure 1 using the default values $\beta = \frac{2}{3}$, $\gamma = -0.1$, $\zeta = 1.1$ [29].

### 4.4 Magnitude Pruning

MAGNITUDEPRUNING [43, 45] is a straightforward way to induce sparsity by zeroing out small values. Here we specifically refer to the strategy of Zhu and Gupta [45], who eliminate all but the $k$ largest parameters each iteration. Initially $k = d$, and it is decayed as a function of iteration number until it reaches the desired sparsity level according to a general-purpose pruning schedule. This is equivalent to IHT [7] (except with decreasing $k$) and therefore almost surely corresponds to a proximal gradient variant of adaptive dropout where $\widehat{\alpha}_j(\mathbf{v}) = \mathbb{1}\{j \in \text{TOP-}k(\mathbf{w})\}$, or $\widehat{\eta}_j(\mathbf{v}) = \infty\mathbb{1}\{j \in \text{TOP-}k(\mathbf{v})\}$, implemented as in Appendix B.

According to Table 1, the effective penalty is the $\text{HARDTHRESH}(k)$ penalty, which we plot in Figure 1. On a philosophical note, this is by definition the "correct" penalty to use when searching for solutions with a constraint on the number of nonzeros, since it is the characteristic function of that set of solutions. The $\eta$-trick along with alternating optimization turns this combinatorial search problem into a sequence of tractable gradient-based (for $\mathbf{w}$) and closed-form (for $\boldsymbol{\eta}$) updates. It is therefore no surprise that Gale et al. [15] observed that this "simple" magnitude pruning algorithm matched or outmatched the other more sophisticated methods.

## 5 Empirical Comparison for VARIATIONALDROPOUT

With complex methods such as the adaptive dropout methods we discuss above, it is often difficult to isolate what drives their success. Under our theory, such methods should be able to be decomposed into dropout noise-related effects and $\eta$-trick effects. Diving further into the VARIATIONALDROPOUT algorithm, we compare it against other $\eta$-trick strategies, examining the effects of dropout noise, $\eta$-trick optimization strategy, and regularization with the underlying penalty.

We apply the methods on the task of sparsifying a LeNet-300-100 network on the MNIST dataset, a common benchmark for sparsification methods [17, 16, 12]. Experimental details along with layer-wise sparsification results are given in Appendix C. We compare the following methods. Except for VARIATIONALDROPOUT methods, we use the LOGSUM penalty.

- **VARDROP+LR+AR.** VARIATIONALDROPOUT as proposed by Molchanov et al. [32], using both the local reparameterization trick and the additive reparameterization trick.
- **VARDROP+LR.** VARIATIONALDROPOUT, using the local reparameterization trick but not the additive reparameterization trick.
- **VARDROP.** VARIATIONALDROPOUT with no reparameterization tricks.
- **$\eta$-TRICK.** Joint gradient descent in $\mathbf{w}$ and $\log(\boldsymbol{\eta})$ of (6). This would be equivalent to VARIATIONALDROPOUT with no noise in linear regression.
- **ADAPROX.** Proximal gradient descent in $\mathbf{w}$ of (6) using $\boldsymbol{\eta} = \widehat{\boldsymbol{\eta}}(\mathbf{w})$ at each iteration.
- **ADATIKHONOV.** Gradient descent in $\mathbf{w}$ of (6) using $\boldsymbol{\eta} = \widehat{\boldsymbol{\eta}}(\mathbf{w})$ at each iteration.
- **LOGSUM.** Gradient descent in $\mathbf{w}$ of (1) directly.

We plot the results in Figure 2. With all of the methods compared, we see similar behavior: the networks become more sparse over time as well as more accurate. In general, the methods appear to exhibit a sparsity–accuracy trade-off, even when they optimize identical objective functions (consider

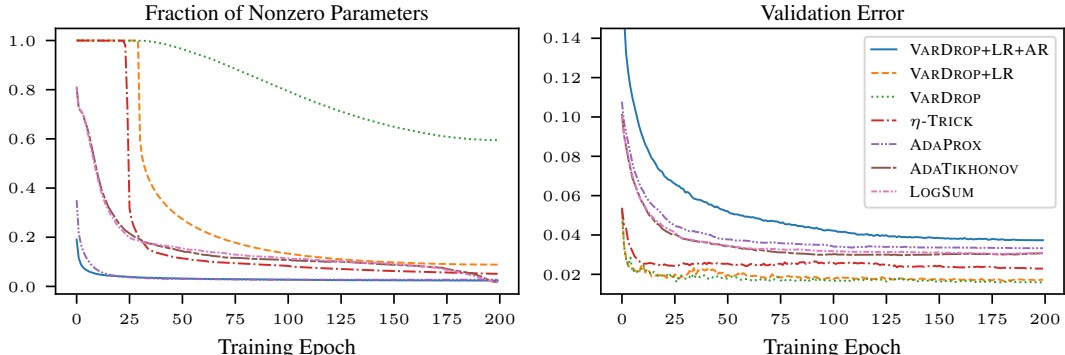

Figure 2: VARIATIONALDROPOUT compared with other $\eta$-trick strategies using a LOGSUM penalty.

VARDROP+LR+AR and VARDROP+LR, or ADAPROX and $\eta$-TRICK). This trade-off seems to be determined by the early iterations, depending on whether the method sparsifies the network quickly or not. We also observe that the stochastic noise due to dropout in VARDROP leads to very slow convergence, while the variance-reduced VARDROP+LR converges exceedingly similarly to its non-random counterpart $\eta$-TRICK. In addition, VARDROP+LR+AR behaves remarkably similarly to ADAPROX, which was inspired by the additive reparameterization trick (see discussion on reparameterization tricks in Section 4). These results suggests that the success of VARIATIONALDROPOUT is well explained by interpreting it as optimization of a dual formulation of a regularized ERM problem. Further, they suggest that the variance due to dropout noise at best yields similar performance to the non-dropout counterpart and at worst can drastically slow down convergence. Interestingly, the simple LOGSUM-regularized ERM approach performs as well as any of the other methods considered, and is nearly identical in behavior to ADATIKHONOV, suggesting that the sparsification performance of VARIATIONALDROPOUT is entirely due to the effective LOGSUM penalty.

## 6    Discussion

Given the duality between adaptive dropout and regularization that we have presented, it is no surprise that such methods excel at sparsifying deep networks. However, many questions still remain.

For one, is there any benefit to dropout noise itself in adaptive dropout optimization, beyond the $\eta$-trick connection? There are known benefits of standard dropout for deep network optimization [1, 19, 31, 41], but do these benefits transfer to adaptive dropout strategies? A crucial component of VARIATIONALDROPOUT, as demonstrated in Section 5, appears to be that its successful implementations *remove* most of the noise due to dropout. On the other hand, methods like HARD-CONCRETEL0NORM from Louizos et al. [29] exploit the fact that sparse masks yield more efficient computation, and they do not seem to suffer from variance-related optimization issues even without taking efforts to mitigate such effects. This could be because they use a biased dropout distribution, which results in significantly lower variance of stochastic gradients. On the other hand, it is not clear that dropout noise is necessary to obtain these computational speedups; other methods where the network is sparse during training, such as MAGNITUDEPRUNING, should have the similar properties.

It is evident from Figure 2 that, even when using the same objective function, the optimization strategy can have a significant impact on the result. It is beyond the scope of this work to examine the convergence rates, even for convex problems, of the adaptive algorithms we consider in Section 5, although many of them can be cast as stochastic majorization-minimization (MM) algorithms, for which some theory exists [30]. However, for non-convex penalties and losses, such as in sparse deep network optimization, there are no guarantees about the quality of the solution. It is also not clear what benefit $\eta$-trick optimization provides in this setting, if any. For linear problems it has the advantage of drastically simplifying implementations of penalized methods by reducing them to successive least squares problems, but with modern automatic differentiation packages it is straightforward to implement any differentiable sparsity-inducing penalty. To the best of our knowledge, the only large-scale comparison of sparsity methods for deep networks is that of Gale et al. [15], and they do not consider classical penalty-based methods.

We have fully analyzed the adaptive-dropout–regularization duality in the linear regression case, and we know that the exact correspondence between dropout and Tikhonov regularization does not hold

in other settings, such as deep networks. However, from the neural tangent kernel (NTK) [23, 28] perspective, very wide networks become linearized, and parameter-level dropout becomes dropout for linear regression, like we study. Furthermore, recent work has shown that for very high dimensional settings, the choice between classification and regression loss functions is often irrelevant [33, 21]. For finite-width settings, good quadratic approximations to the general effective regularization of dropout have been developed [39, 41], which relate to the Fisher information. These approximations could be used to develop adaptive sparsification strategies that enjoy the data-sensitive properties of dropout, such as in logistic regression or deep networks.

The duality presented in this paper can be the basis of developing more efficient principled sparsification schemes as future work. In particular, using penalties that have been proposed in the compressed sensing literature such as scaled lasso [37], which is adaptive to noise level, or SLOPE [36], which is shown to be minimax optimal in feature selection, can lead to adaptive dropout schemes with better performance in sparsifying deep networks. In addition, analytical tools such as debiasing [25], which has been used for near-optimal feature selection [24], could be employed to design improved sparsity methods.

## Acknowledgments and Disclosure of Funding

This work was supported by NSF grants CCF-1911094, IIS-1838177, and IIS-1730574; ONR grants N00014-18-12571, N00014-20-1-2534, and MURI N00014-20-1-2787; AFOSR grant FA9550-18-1-0478; and a Vannevar Bush Faculty Fellowship, ONR grant N00014-18-1-2047.

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
