# A Appendix: Dual Forms of Subquadratic Penalties

## A.1 Dual Forms for Common Penalties

The full version of the Table 1 is given in Table 3. The dual formulations are derived in the next sections.

Table 3: Common penalties and their corresponding dual formulations.

| Penalty | $\Omega(\mathbf{w})$ | $f(\boldsymbol{\eta})$ | $\widehat{\eta}_j(\mathbf{w})$ |
|---|---|---|---|
| $\ell_1$ | $\lvert w_j \rvert$ | $\eta_j$ | $\lvert w_j \rvert$ |
| $\ell_p,\ p \in (0,2)$ | $\lVert \mathbf{w} \rVert_p$ | $\lVert \boldsymbol{\eta} \rVert_q : q = \frac{p}{2-p}$ | $\lvert w_j \rvert^{2-p} \lVert \mathbf{w} \rVert_p^{p-1}$ |
| $\ell_p^p,\ p \in (0,2)$ | $\frac{1}{p} \lvert w_j \rvert^p$ | $\frac{1}{q} \eta_j^q : q = \frac{p}{2-p}$ | $\lvert w_j \rvert^{2-p}$ |
| $\ell_0$ | $\mathbb{1}\{\lvert w_j \rvert > 0\}$ | $2\mathbb{1}\{\eta_j > 0\}$ | $\infty \mathbb{1}\{\lvert w_j \rvert > 0\}$ |
| $\textsc{ElasticNet}(\theta)$ [46] | $\frac{\theta}{2} w_j^2 + (1-\theta)\lvert w_j \rvert$ | $\frac{\eta_j(1-\theta)^2}{1-\eta_j\theta},\ \mathcal{H} = [0, \frac{1}{\theta}]$ | $\frac{\lvert w_j \rvert}{\lvert w_j \rvert \theta + (1-\theta)}$ |
| $\textsc{Huber}(\varepsilon)$ [11, 22] | $\begin{cases} \frac{1}{2\varepsilon} w_j^2 + \frac{\varepsilon}{2}, & \lvert w_j \rvert \leqslant \varepsilon \\ \lvert w_j \rvert, & \lvert w_j \rvert > \varepsilon \end{cases}$ | $\eta_j,\ \mathcal{H} = [\varepsilon, \infty)$ | $\max\{\varepsilon, \lvert w_j \rvert\}$ |
| $\textsc{LogSum}(\varepsilon)$ [9] | $\log(\lvert w_j \rvert + \varepsilon)$ | $2\log\left(\frac{\sqrt{\varepsilon^2 + 4\eta_j} + \varepsilon}{2}\right) - \frac{\left(\sqrt{\varepsilon^2 + 4\eta_j} - \varepsilon\right)^2}{4\eta_j}$ | $\lvert w_j \rvert(\lvert w_j \rvert + \varepsilon)$ |
| $\textsc{SCAD}(a, \lambda)$ [13] | $\begin{cases} \lvert w_j \rvert, & \lvert w_j \rvert \leqslant \lambda \\ \frac{2a\lambda\lvert w_j \rvert - w_j^2 - \lambda^2}{2(a-1)\lambda}, & \lvert w_j \rvert \in (\lambda, a\lambda] \\ \frac{(a+1)\lambda}{2}, & \lvert w_j \rvert > a\lambda \end{cases}$ | $\begin{cases} \eta, & \eta_j \leqslant \lambda \\ \lambda\frac{(a+1)\eta_j - \lambda}{(a-1)\lambda + \eta_j}, & \eta_j > \lambda \end{cases}$ | $\begin{cases} \lvert w_j \rvert, & \lvert w_j \rvert \leqslant \lambda \\ \frac{(a-1)\lambda\lvert w_j \rvert}{a\lambda - \lvert w_j \rvert}, & \lvert w_j \rvert \in (\lambda, a\lambda] \\ \infty, & \lvert w_j \rvert > a\lambda \end{cases}$ |
| $\textsc{MCP}(a, \lambda)$ [44] | $\begin{cases} \lvert w_j \rvert - \frac{w_j^2}{2a\lambda}, & \lvert w_j \rvert \leqslant a\lambda \\ \frac{a\lambda}{2}, & \lvert w_j \rvert > a\lambda \end{cases}$ | $\frac{a\lambda\eta_j}{\eta_j + a\lambda}$ | $\begin{cases} \frac{a\lambda\lvert w_j \rvert}{a\lambda - \lvert w_j \rvert}, & \lvert w_j \rvert < a\lambda \\ \infty, & \lvert w_j \rvert \geqslant a\lambda \end{cases}$ |
| $\textsc{HardThresh}(k)$ [7] | $\infty\mathbb{1}\{\lVert \mathbf{w} \rVert_0 > k\}$ | $0,\ \mathcal{H} = \{\boldsymbol{\eta} : \lVert \boldsymbol{\eta} \rVert_0 \leqslant k\}$ | $\infty\mathbb{1}\{j \in \textsc{Top-}k(\mathbf{w})\}$ |

## A.2 General Strategy

Define $g(\mathbf{u}) := \Omega(\mathbf{u}^{\odot\frac{1}{2}})$ for $\mathbf{u} \in \mathbb{R}_+^d$ and $h(\mathbf{v}) := f(-\mathbf{v}^{\odot-1})$ for $\mathbf{v} \in \overline{\mathbb{R}}_{\leqslant 0}^d$. As mentioned in Section 2.2, when $g$ is concave, $-2g$ and $h$ comprise a Legendre–Fenchel conjugate pair, each being convex functions. That is, the following relationships hold:

$$-2g(\mathbf{u}) = \sup_{\mathbf{v}} \mathbf{u}^\top \mathbf{v} - h(\mathbf{v}), \qquad h(\mathbf{v}) = \sup_{\mathbf{u}} \mathbf{u}^\top \mathbf{v} + 2g(\mathbf{u}). \tag{22}$$

We can thus obtain $-2g$ and $h$ from each other by solving these optimizations. If the functions are differentiable, the following first-order conditions must hold for the dual pair $\mathbf{u}^*$ and $\mathbf{v}^*$, the argument and maximizing variable in either equation in (22):

$$\mathbf{u}^* = \nabla_{\mathbf{v}} h(\mathbf{v}^*), \qquad \mathbf{v}^* = -2\nabla_{\mathbf{u}} g(\mathbf{u}^*). \tag{23}$$

Note that the second condition is equivalent to (7). Once we have characterized $g$ and $h$, we can recover $\Omega$ and $f$ by considering $\mathbf{w}^{\odot 2} = \mathbf{u}$ and $\boldsymbol{\eta} = -\mathbf{v}^{\odot-1}$. In the following sections, we use these properties to obtain the dual forms presented in Table 1. For separable penalties, it suffices to derive the dual form of the scalar penalty.

## A.3 $\ell_p$ for $0 < p < 2$

This formulation can be found in Lemma 3.1 of Jenatton et al. [26], but we present another derivation here. First we compute the gradient

$$g(\mathbf{u}) = \left(\sum_j u_j^{\frac{p}{2}}\right)^{\frac{1}{p}} \implies \nabla_{\mathbf{u}} g(\mathbf{u}) = \frac{1}{2}\mathbf{u}^{\odot\frac{p}{2}-1}\left(\sum_j u_j^{\frac{p}{2}}\right)^{\frac{1}{p}-1}. \tag{24}$$

The the first-order condition is

$$\mathbf{v}^* = -\mathbf{u}^{*\odot\frac{p-2}{2}} g(\mathbf{u}^*)^{1-p}, \tag{25}$$

which gives $\widehat{\eta}_j(\mathbf{w}) = |w_j|^{2-p}\|\mathbf{w}\|_p^{p-1}$. Now then

$$h(\mathbf{v}^*) = -\left(\sum_j u_j^{*\frac{p}{2}}\right) g(\mathbf{u}^*)^{1-p} + 2g(\mathbf{u}^*) \tag{26}$$

$$= -g(\mathbf{u}^*)^p g(\mathbf{u}^*)^{1-p} + 2g(\mathbf{u}^*) \tag{27}$$

$$= g(\mathbf{u}^*). \tag{28}$$

Since $g(a\mathbf{z}) = \sqrt{a}g(\mathbf{z})$, we can solve (25) for $g(\mathbf{u}^*)$:

$$g(\mathbf{u}^*) = g\left(\left(\frac{-\mathbf{v}^*}{g(\mathbf{u}^*)^{1-p}}\right)^{\odot\frac{2}{p-2}}\right) \tag{29}$$

$$= g(\mathbf{u}^*)^{\frac{p-1}{p-2}} g\left((-\mathbf{v}^*)^{\odot\frac{2}{p-2}}\right) \tag{30}$$

$$\implies g(\mathbf{u}^*)^{\frac{1}{2-p}} = \left(\sum_j \left(-\frac{1}{v_j}\right)^{\frac{p}{2-p}}\right)^{\frac{1}{p}} \tag{31}$$

$$\implies h(\mathbf{v}^*) = \left(\sum_j \left(-\frac{1}{v_j}\right)^{\frac{p}{2-p}}\right)^{\frac{2-p}{p}}. \tag{32}$$

Thus, $f(\boldsymbol{\eta}) = \|\boldsymbol{\eta}\|_q$ for $q = \frac{p}{2-p}$.

## A.4  $\ell_p^p$ for $0 < p < 2$

First we compute the derivative

$$g(u) = \frac{1}{p}u^{\frac{p}{2}} \quad\implies\quad g'(u) = \frac{1}{2}u^{\frac{p}{2}-1}. \tag{33}$$

Then the first-order condition is

$$v^* = -u^{*\frac{p-2}{2}}, \tag{34}$$

which gives $\widehat{\eta}(w) = |w|^{2-p}$. We also have $u^* = -v^{*\frac{2}{p-2}}$, which gives us

$$h(v^*) = -(-v^*)^{\frac{2}{p-2}+1} + \frac{2}{p}(-v^*)^{\frac{p}{p-2}} \tag{35}$$

$$= \frac{2-p}{p}\left(-\frac{1}{v^*}\right)^{\frac{p}{2-p}}. \tag{36}$$

Thus, $f(\eta) = \frac{1}{q}\eta^q$ for $q = \frac{p}{2-p}$.

## A.5  Elastic Net

First, we compute the derivative

$$g(u) = \frac{\theta}{2}u + (1-\theta)\sqrt{u} \quad\implies\quad g'(u) = \frac{\theta}{2} + \frac{1-\theta}{2\sqrt{u}}. \tag{37}$$

The first-order condition is

$$v^* = -\theta - \frac{1-\theta}{\sqrt{u^*}}, \tag{38}$$

which is bounded by $v^* \leqslant -\theta$. From this we obtain $\widehat{\eta}(w) = \frac{|w|}{|w|\theta+(1-\theta)}$. We also have $\sqrt{u^*} = \frac{1-\theta}{-v^*-\theta}$. This gives us

$$h(v^*) = \frac{v^*(1-\theta)^2}{(-v^*-\theta)^2} + \frac{\theta(1-\theta)^2}{(-v^*-\theta)^2} + \frac{2(1-\theta)^2}{(-v^*-\theta)} \tag{39}$$

$$= \frac{(1-\theta)^2}{(-v^*-\theta)}. \tag{40}$$

Thus, $f(\eta) = \frac{\eta(1-\theta)^2}{1-\eta\theta}$ for $\eta \leqslant \frac{1}{\theta}$.

## A.6 Huber

As usual, first we compute the derivative

$$g(u) = \begin{cases} \frac{1}{2\varepsilon}u + \frac{\varepsilon}{2}, & \sqrt{u} \leqslant \varepsilon \\ \sqrt{u}, & \sqrt{u} > \varepsilon \end{cases} \implies g'(u) = \begin{cases} \frac{1}{2\varepsilon}, & \sqrt{u} \leqslant \varepsilon \\ \frac{1}{2\sqrt{u}}, & \sqrt{u} > \varepsilon \end{cases}. \tag{41}$$

The first-order condition is

$$v^* = -\min\left\{\frac{1}{\varepsilon}, \frac{1}{\sqrt{u^*}}\right\}, \tag{42}$$

which is bounded by $v^* \geqslant -\frac{1}{\varepsilon}$. This gives us $\widehat{\eta}(w) = \max\{\varepsilon, |w|\}$. For $v^* \geqslant -\frac{1}{\varepsilon}$, $\sqrt{u^*} = -\frac{1}{v^*}$, so

$$h(v^*) = \frac{1}{v^*} - 2\frac{1}{v^*} = -\frac{1}{v^*}. \tag{43}$$

Thus, $f(\eta) = \eta$ for $\eta \geqslant \varepsilon$.

## A.7 Log Sum

First, we compute the derivative

$$g(u) = \log(\sqrt{u} + \varepsilon) \implies g'(u) = \frac{1}{2\sqrt{u}(\sqrt{u} + \varepsilon)}. \tag{44}$$

Then the first-order condition is

$$v^* = -\frac{1}{\sqrt{u^*}(\sqrt{u^*} + \varepsilon)}. \tag{45}$$

This gives us $\widehat{\eta}(w) = |w|(|w| + \varepsilon)$. Rewriting the above as a quadratic equation in $\sqrt{u^*}$, we have

$$(\sqrt{u^*})^2 + \varepsilon\sqrt{u^*} + \frac{1}{v^*} = 0, \tag{46}$$

which gives the inverse mapping $\sqrt{u^*} = \frac{\sqrt{\varepsilon^2 - \frac{4}{v^*}} - \varepsilon}{2}$. Thus we get

$$h(v^*) = \frac{v^*}{4}\left(\sqrt{\varepsilon^2 - \frac{4}{v^*}} - \varepsilon\right)^2 + 2\log\left(\frac{\sqrt{\varepsilon^2 - \frac{4}{v^*}} + \varepsilon}{2}\right). \tag{47}$$

Thus, $f(\eta) = 2\log\left(\frac{\sqrt{\varepsilon^2 + 4\eta} + \varepsilon}{2}\right) - \frac{1}{4\eta}\left(\sqrt{\varepsilon^2 + 4\eta} - \varepsilon\right)^2$.

## A.8 SCAD

The SCAD penalty as presented by Fan and Li [13] uses the regularization scaling $\lambda$ as a parameter, so first we factor it out:

$$\lambda\Omega(w) = \lambda \begin{cases} |w|, & |w| \leqslant \lambda \\ \frac{2a\lambda|w| - w^2 - \lambda^2}{2(a-1)\lambda}, & |w| \in (\lambda, a\lambda] \\ \frac{(a+1)\lambda}{2}, & |w| > a\lambda \end{cases}. \tag{48}$$

We then compute the derivative

$$g(u) = \begin{cases} \sqrt{u}, & \sqrt{u} \leqslant \lambda \\ \frac{2a\lambda\sqrt{u} - u - \lambda^2}{2(a-1)\lambda}, & \sqrt{u} \in (\lambda, a\lambda] \\ \frac{(a+1)\lambda}{2}, & \sqrt{u} > a\lambda \end{cases} \implies g'(u) = \begin{cases} \frac{1}{2\sqrt{u}}, & \sqrt{u} \leqslant \lambda \\ \frac{a}{2(a-1)\sqrt{u}} - \frac{1}{2(a-1)\lambda}, & \sqrt{u} \in (\lambda, a\lambda] \\ 0, & \sqrt{u} > a\lambda \end{cases}. \tag{49}$$

This gives us the first order condition and in turn $\widehat{\eta}$:

$$v^* = \begin{cases} -\frac{1}{\sqrt{u^*}}, & \sqrt{u^*} \leqslant \lambda \\ -\frac{a}{(a-1)\sqrt{u^*}} + \frac{1}{(a-1)\lambda}, & \sqrt{u^*} \in (\lambda, a\lambda] \\ 0, & \sqrt{u^*} > a\lambda \end{cases} \tag{50}$$

$$\implies \widehat{\eta}(w) = \begin{cases} |w|, & |w| \leqslant \lambda \\ \frac{(a-1)\lambda|w|}{a\lambda - |w|}, & |w| \in (\lambda, a\lambda] \\ \infty, & |w| > a\lambda \end{cases}. \tag{51}$$

Now when $\sqrt{u^*} \leqslant \lambda$, $v^* \leqslant -\frac{1}{\lambda}$, and when $\sqrt{u^*} \in (\lambda, a\lambda]$, $v^* \in (-\frac{1}{\lambda}, 0]$. In the first case, $\sqrt{u^*} = -\frac{1}{v^*}$, and in the second, $\sqrt{u^*} = \frac{a\lambda}{1-(a-1)\lambda v^*}$. Therefore,

$$h(v^*) = \begin{cases} \frac{1}{v^*} - \frac{2}{v^*}, & v^* \leqslant -\frac{1}{\lambda} \\ \frac{a^2\lambda^2 v^*}{(1-(a-1)\lambda v^*)^2} + \frac{2a\lambda\left(\frac{a\lambda}{1-(a-1)\lambda v^*}\right) - \frac{a^2\lambda^2}{(1-(a-1)\lambda v^*)^2} - \lambda^2}{(a-1)\lambda}, & v^* > -\frac{1}{\lambda} \end{cases} \tag{52}$$

$$= \begin{cases} -\frac{1}{v^*}, & v^* \leqslant -\frac{1}{\lambda} \\ \frac{a^2(a-1)\lambda^3 v^* + 2a^2\lambda^2(1-(a-1)\lambda v^*) - a^2\lambda^2 - \lambda^2(1-(a-1)\lambda v^*)^2}{(a-1)\lambda(1-(a-1)\lambda v^*)^2}, & v^* > -\frac{1}{\lambda} \end{cases} \tag{53}$$

$$= \begin{cases} -\frac{1}{v^*}, & v^* \leqslant -\frac{1}{\lambda} \\ \frac{\lambda(a^2 - a^2(a-1)\lambda v^* - (1-(a-1)\lambda v^*)^2)}{(a-1)(1-(a-1)\lambda v^*)^2}, & v^* > -\frac{1}{\lambda} \end{cases} \tag{54}$$

$$= \begin{cases} -\frac{1}{v^*}, & v^* \leqslant -\frac{1}{\lambda} \\ \frac{\lambda(a^2 - 1 + (a-1)\lambda v^*)}{(a-1)(1-(a-1)\lambda v^*)}, & v^* > -\frac{1}{\lambda} \end{cases} \tag{55}$$

$$= \begin{cases} -\frac{1}{v^*}, & v^* \leqslant -\frac{1}{\lambda} \\ \frac{\lambda(a+1+\lambda v^*)}{1-(a-1)\lambda v^*}, & v^* > -\frac{1}{\lambda} \end{cases}. \tag{56}$$

From this we obtain

$$f(\eta) = \begin{cases} \eta, & \eta \leqslant \lambda \\ \lambda\frac{(a+1)\eta - \lambda}{(a-1)\lambda + \eta}, & \eta > \lambda. \end{cases} \tag{57}$$

## A.9 MCP

As with SCAD, we first factor out the $\lambda$ from the penalty:

$$\lambda\Omega(w) = \lambda \begin{cases} |w| - \frac{w^2}{2a\lambda}, & |w| \leqslant a\lambda \\ \frac{a\lambda}{2}, & |w| > a\lambda \end{cases}. \tag{58}$$

We then compute the derivative

$$g(u) = \begin{cases} \sqrt{u} - \frac{u}{2a\lambda}, & \sqrt{u} \leqslant a\lambda \\ \frac{a\lambda}{2}, & \sqrt{u} > a\lambda \end{cases} \implies g'(u) = \begin{cases} \frac{1}{2\sqrt{u}} - \frac{1}{2a\lambda}, & \sqrt{u} \leqslant a\lambda \\ 0, & \sqrt{u} > a\lambda \end{cases}. \tag{59}$$

Our first-order condition is

$$v^* = \begin{cases} -\frac{1}{\sqrt{u^*}} + \frac{1}{a\lambda}, & \sqrt{u^*} \leqslant a\lambda \\ 0, & \sqrt{u^*} > a\lambda \end{cases}, \tag{60}$$

from which we obtain

$$\widehat{\eta}(w) = \begin{cases} \frac{a\lambda|w|}{a\lambda - |w|}, & |w| < a\lambda \\ \infty, & |w| \geqslant a\lambda \end{cases}. \tag{61}$$

We have the inverse mapping $\sqrt{u^*} = \frac{a\lambda}{1-a\lambda v^*}$, which gives us

$$h(v^*) = \frac{a^2\lambda^2 v^*}{(1-a\lambda v^*)^2} + \frac{2a\lambda}{1-a\lambda v^*} - \frac{a\lambda}{(1-a\lambda v^*)^2} \tag{62}$$

$$= \frac{a\lambda(a\lambda v^* + 2(1-a\lambda v^*) - 1)}{(1-a\lambda v^*)^2} \tag{63}$$

$$= \frac{a\lambda}{1-a\lambda v^*}. \tag{64}$$

From here, we directly obtain $f(\eta) = \frac{a\lambda\eta}{\eta + a\lambda}$.

## A.10 $\quad \ell_0$

The $\ell_0$ penalty is not differentiable. However, it is separable, and in one dimension we have

$$g(u) = \mathbb{1}\{u > 0\}. \tag{65}$$

Thus $-2g$ is convex since its epigraph is a convex set. For $u = 0$, $-2g$ has a supporting line with slope $-\infty$, and elsewhere with slope $0$. Thus we have the relationship $v^* = -\infty\mathbb{1}\{u^* = 0\}$, which yields $\widehat{\eta}(w) = \infty\mathbb{1}\{|w| > 0\}$. The mapping $u^* \mapsto v^*$ is not invertible, so we consider two cases of $v^*$:

$$h(v^*) = \begin{cases} 0, & v^* = -\infty \\ \sup_{u>0} uv^* + 2g(u), & v^* > -\infty \end{cases} \tag{66}$$

$$= 2\mathbb{1}\{v^* > -\infty\}. \tag{67}$$

We thus conclude that $f(\eta) = \mathbb{1}\{\eta > 0\}$.

## A.11 Hard Threshold

For this penalty, we begin with the $\widehat{\eta}(\mathbf{w})$ that yields the IHT algorithm when $\mathbf{w}$ is optimized by a gradient step. This corresponds to

$$\widehat{\eta}_j(\mathbf{w}) = \infty\mathbb{1}\{j \in \text{Top-}k(\mathbf{w})\}. \tag{68}$$

We thus seek to find a penalty that yields such an $\widehat{\boldsymbol{\eta}}$. In interest of mathematical preciseness, let us define, given $a > 0$ and $m \in [d]$, the set

$$\mathcal{S}_{-a}^m := \text{Conv}\left(\{\mathbf{v} : v_j \in \{-a, 0\}, \#\{j : v_j = -a\} \geqslant m\}\right), \tag{69}$$

where $\text{Conv}(\mathcal{A})$ is the convex hull of the set $\mathcal{A}$. Similarly define

$$\bar{\mathcal{S}}_{-a}^m := \text{Conv}\left(\{\mathbf{v} : v_j \in [-\infty, -a] \cup \{0\}, \#\{j : v_j \leqslant -a\} \geqslant m\}\right), \tag{70}$$

and lastly define

$$\widehat{\mathcal{S}}_{-a}^m := \{\mathbf{v} : v_j \leqslant v_j' \ \forall j \text{ for some } \mathbf{v}' \in \mathcal{S}_{-a}^m\}. \tag{71}$$

Note that $\mathcal{S}_{-a}^m \subseteq \bar{\mathcal{S}}_{-a}^m \subseteq \widehat{\mathcal{S}}_{-a}^m$ and that $\widehat{\mathcal{S}}_{-a}^m$ is also a convex set. Now consider

$$h_a(\mathbf{v}) = \infty\mathbb{1}\left\{\mathbf{v} \notin \bar{\mathcal{S}}_{-a}^{d-k}\right\}. \tag{72}$$

This function is convex as it has a convex epigraph. Its Legendre–Fenchel transform is given by

$$h_a^*(\mathbf{u}) = \sup_{\mathbf{v}} \mathbf{u}^\top \mathbf{v} - h_a(\mathbf{v}) \tag{73}$$

$$= \sup_{\mathbf{v} \in \bar{\mathcal{S}}_{-a}^{d-k}} \mathbf{u}^\top \mathbf{v} \tag{74}$$

$$\leqslant \sup_{\mathbf{v} \in \widehat{\mathcal{S}}_{-a}^{d-k}} \mathbf{u}^\top \mathbf{v} \tag{75}$$

$$= \sup_{\mathbf{v} \in \mathcal{S}_{-a}^{d-k}} \mathbf{u}^\top \mathbf{v} \quad, \tag{76}$$

where inequality holds because $\bar{\mathcal{S}}_{-a}^{d-k} \subseteq \hat{\mathcal{S}}_{-a}^{d-k}$ the final equality holds by definition of $\hat{\mathcal{S}}_{-a}^{d-k}$. Clearly, the inequality is equality since $\mathcal{S}_{-a}^{d-k} \subseteq \bar{\mathcal{S}}_{-a}^{d-k}$. Now consider that for any $\mathbf{v} \in \mathcal{S}_{-a}^{d-k}$, $\sum_j v_j \leqslant -(d-k)a$ and $v_j \geqslant -a \ \forall j$. We can choose at most $k$ elements of $\mathbf{v}$ to be zero, so to achieve the supremum we must choose them at the largest elements of $\mathbf{u}$. That leaves then that the remaining elements must be $-a$, so we have

$$h_a^*(\mathbf{u}) = -a \sum_{j>k} u_{(j)}. \tag{77}$$

With corresponding $v_j^* = -a\mathbb{1}\{j \notin \text{TOP-}k(\mathbf{u}^*)\}$. Now, taking $a \to \infty$ for $\mathbf{v}^*$, $h_a$, and $h_a^*$ we can determine $\boldsymbol{\eta}$, $f$, and $\Omega$. First, as desired,

$$\boldsymbol{\eta}_j(\mathbf{w}) = \lim_{a\to\infty} - (-a\mathbb{1}\{j \notin \text{TOP-}k(\mathbf{w})\})^{-1} \tag{78}$$

$$= \infty\mathbb{1}\{j \in \text{TOP-}k(\mathbf{w})\}. \tag{79}$$

Then, since $h_a(\mathbf{v})$ is infinite for $\mathbf{v} \notin \bar{\mathcal{S}}_{-a}^{d-k}$ and zero for $\mathbf{v} \in \bar{\mathcal{S}}_{-a}^{d-k}$, we have $f(\boldsymbol{\eta}) = 0$ with

$$\mathcal{H} = \lim_{a\to\infty} \left\{ \boldsymbol{\eta} : -\boldsymbol{\eta}^{\odot-1} \in \bar{\mathcal{S}}_{-a}^{d-k} \right\} \tag{80}$$

$$= \left\{ \boldsymbol{\eta} : \|\boldsymbol{\eta}\|_0 \leqslant k \right\}. \tag{81}$$

Lastly, we have

$$\Omega(\mathbf{w}) = \lim_{a\to\infty} -2h_a^*(\mathbf{w}^{\odot 2}) \tag{82}$$

$$= \infty\mathbb{1}\{\|\mathbf{w}\|_0 > 0\}. \tag{83}$$

## B  Adaptive Dropout with Additive Reparameterization

In Algorithm 1 we present one scheme for implementing adaptive dropout using an additive reparameterization via a two-pass proximal update of the variables $\mathbf{w}$ and $\mathbf{v}$. This method is equivalent to an adaptive proximal stochastic gradient descent with the adaptive Tikhonov penalty.

---

**Algorithm 1:** Adaptive Dropout with Additive Reparameterization

---

**Input:** Differentiable $\mathcal{L} \colon \mathbb{R}^d \to \mathbb{R}$, $\hat{\boldsymbol{\eta}} \colon \mathbb{R}^d \to \mathcal{H}$, $\lambda > 0$, $(\rho_t)_{t=1}^T$, $\mathbf{w}^0$, $\boldsymbol{\alpha}^0$.
**Output:** $\mathbf{w}^T$.
$\mathbf{w}^{0,2} = \mathbf{w}^0$.
**for** $t = 1, 2, \ldots, T$ **do**
    Draw $\mathbf{s}^t \sim \text{MASK}(\boldsymbol{\alpha}^{t-1})$.
    $\mathbf{w}^{t,1} = \mathbf{w}^{t-1,2} - \rho_t \nabla_{\mathbf{w}} \mathcal{L}\left(\mathbf{w}^{t-1,2} + (\mathbf{s}^t - \mathbf{1}) \odot \mathbf{v}^{t-1,2}\right)$.
    $\mathbf{v}^{t,1} = \mathbf{w}^{t,1}$.
    $\boldsymbol{\eta}^t = \hat{\boldsymbol{\eta}}(\mathbf{v}^{t,1})$.
    $\mathbf{v}^{t,2} = \left(\rho_t \lambda \text{diag}\left(\boldsymbol{\eta}^t\right)^{-1} + \mathbf{I}\right)^{-1} \mathbf{v}^{t,1}$.
    $\mathbf{w}^{t,2} = \mathbf{v}^{t,2}$.
    $\alpha_j^t = \frac{\eta_j^t}{\eta_j^t + \lambda} \ \forall j \in [d]$.
**end**

---

## C  Experimental Details

We use the PyTorch [34] and skorch [38] libraries to implement deep network methods. On an Nvidia 980 Ti GPU, the experiment runs in about an hour. We randomly divide the MNIST training set into training and validation sets with an 80/20 split. For methods involving optimization in $\log(\boldsymbol{\eta})$, we optimize instead in $\log(\bar{\boldsymbol{\eta}})$ for $\bar{\boldsymbol{\eta}} = \boldsymbol{\eta}/\lambda$, as Molchanov et al. [32] do. We initialize with $\log(\bar{\eta}_j) = 5$. For the VARDROP methods, we use the dual penalty $f(\bar{\boldsymbol{\eta}})$ and implement the methods using code provided by the authors [2]. For other methods, we simply use the LOGSUM(2) penalty (based on Figure 1) applied to $\boldsymbol{\eta}$ directly, along with a larger value of $\lambda$ to account for the implicit attenuation of the Tikhonov regularization due to dropout with the cross-entropy loss. For all methods, we use the Adam optimizer with a linear decay to 0 of the initial learning rate. The initial learning rate is set

to be $10^{-4}$, but for a few methods this failed to converge to a sparse solution, so we increased it to $10^{-3}$. For VARDROP, convergence was quite slow; running for a longer number of epochs, however, does continue to improve the sparsity. Running for 1000 epochs, for example, gets the fraction of nonzeros down to around 0.1, at a slight expense of accuracy. We report hyperparameters and test error in Table 4.

We measure sparsity using the same method as Molchanov et al. [32]: we count the values of $\bar{\boldsymbol{\eta}}$ such that $\sigma(\bar{\eta}_j) < 0.05$, and we zero out the corresponding $w_j$ when applying the network to a validation/test sample. For $\eta$-TRICK, we observed that while the parameters $\mathbf{w}$ were indeed converging to sparse solutions, the $\boldsymbol{\eta}$ parameters were not, resulting in a mismatch of the actual sparsity of the network and our reported score; to remedy this, we apply a very small penalty of $\lambda \cdot 10^{-3} \log(\bar{\eta})$, which did not seem to compromise network accuracy. We report the fraction of nonzeros for each layer in Table 5.

Table 4: Hyperparameters and final results for sparsification of LeNet-300-100.

| Method | $\lambda$ | Learning Rate | Test Error | Fraction of Nonzeros |
|---|---|---|---|---|
| VARDROP+LR+AR | $\frac{1}{60,000}$ | $10^{-4}$ | 3.21% | 0.024 |
| VARDROP+LR | $\frac{1}{60,000}$ | $10^{-3}$ | 1.41% | 0.088 |
| VARDROP | $\frac{1}{60,000}$ | $10^{-3}$ | 1.54% | 0.595 |
| $\eta$-TRICK | $10^{-3}$ | $10^{-3}$ | 2.16% | 0.051 |
| ADAPROX | $10^{-3}$ | $10^{-4}$ | 2.94% | 0.028 |
| ADATIKHONOV | $10^{-3}$ | $10^{-4}$ | 2.88% | 0.018 |
| LOGSUM | $10^{-3}$ | $10^{-4}$ | 2.93% | 0.019 |

Table 5: Layer-wise sparsification results for LeNet-300-100.

| Method | $784 \times 300$ | $300 \times 100$ | $100 \times 1$ | Total |
|---|---|---|---|---|
| VARDROP+LR+AR | 0.020 | 0.035 | 0.502 | 0.024 |
| VARDROP+LR | 0.072 | 0.189 | 0.999 | 0.088 |
| VARDROP | 0.568 | 0.788 | 1.000 | 0.595 |
| $\eta$-TRICK | 0.054 | 0.026 | 0.206 | 0.051 |
| ADAPROX | 0.026 | 0.024 | 0.399 | 0.028 |
| ADATIKHONOV | 0.016 | 0.025 | 0.460 | 0.018 |
| LOGSUM | 0.016 | 0.025 | 0.479 | 0.019 |