# OpenReview forum: "The Flip Side of the Reweighted Coin: Duality of Adaptive Dropout and Regularization"
_NeurIPS.cc/2021/Conference — NeurIPS 2021 Poster_

### Official Review · Reviewer_U2rc · 2021-07-11

**Rating:** 6
**Confidence:** 2

**Summary:**

The paper proposes to use a dual formulation that majorizes the original dropout regularization, and rewrite the original dropout objective as a joint optimization on both the parameter and the auxiliary variable. The authors further show in the linear case any dropout algorithm, if it adapts to the magnitude of the weights monotonically, is equivalent to a regularized ERM with a subquadratic regularizer, which induces sparsity.


**Limitations And Societal Impact:**

the authors adequately addressed the limitations and potential negative societal impact of their work

**Main Review:**

It is nice to build a connection between the adaptive dropout strategy and sparse solution from the perspective of joint optimization and dual formulation. In this way, the dropout strategies are better understood in terms of the regularizers in the objective. It is a step forward towards understanding the effect of dropout.

On the other hand, the analysis of the draft is mostly limited to the linear loss. For deep models, due to its non-linearity, the impact of dropout may not be captured fully as a regularizer in the objective. For example, it is still unclear to me how dropout may interact with the optimization algorithms during the training.

A follow up question: is there any particular reason the authors used the non-linear LeNet model in the experiment? Feels like the analysis is mostly about linear models.


**Time Spent Reviewing:**

2 hours

---

> ### Author Response · Authors · 2021-08-09
> **Authors' Response**
>
> We thank the reviewer for their thoughtful comments.
>
> We agree with the reviewer that there are limitations in considering the linear setting; we refer the reviewer to our other responses where we discuss these limitations in more detail.
>
> Regarding using the non-linear model in the experiment, yes, our intention was to demonstrate that our characterization of the regularization of adaptive dropout algorithms explains results even beyond linear regression, such as in deep networks where algorithms like variational dropout are used. That is, even though the analysis is based on linear models, we believe the applicability of the ideas we present are much broader.

---

### Official Review · Reviewer_8X6v · 2021-07-19

**Rating:** 7
**Confidence:** 3

**Summary:**

This paper draws connections between adaptive dropout and regularized optimization using the $\eta$-trick. The authors show that various adaptive dropout methods can be reinterpreted as sparsity-induced regularization for linear regression models.


**Limitations And Societal Impact:**

The limitation was discussed, and the negative societal impact was not. I did not see any negative society impact neither.

**Main Review:**


I think this work is solid and sound. The main contribution of this paper is that the authors provide a unified framework for analyzing the existing adaptive dropout methods and thus provide theoretical justifications for them. The analysis explains the underlying mechanisms of some adaptive dropout methods, which can potentially contribute to the research in developing strategies for sparse training of neural networks.


1. One concern is that the adaptive dropout methods are not so commonly used in practice (e.g., for training neural networks) and are also more complicated than the Bernoulli dropout. However, it violates the monotonic assumption and thus doesn't fit in your framework.


2. Also, the current analysis is limited to linear regression problems. It may not hold for linear classification problems, for which the closed-form solution may not exist and hence add extra difficulty for the analysis. However, classification problems are more common. Do you think similar conclusions can still be made for linear classification problems with adaptive dropout?


3. In Figure 2, you showed similar behaviors of variational dropout with its dual formulation in terms of the fraction of nonzero parameters and validation error. I wonder if they induce similar sparsity patterns (e.g., the fraction of nonzero parameters at each layer)?


Overall, I am positive about this paper as it provides a unified framework for analyzing adaptive dropout methods with mild assumptions. It provides theoretical justifications for these methods, as some of them are proposed based on heuristics. Therefore, I vote for acceptance.

====Post rebuttal=======

Thanks for the response. My questions & concerns were addressed by the response. I would suggest the authors include the additional empirical results in the revised paper. Though the theory is for linear cases, if the authors can empirically demonstrate that the theory aligns well with non-linear systems, such as neural networks, I think this will make the paper stronger.

**Time Spent Reviewing:**

6

---

> ### Author Response · Authors · 2021-08-09
> **Authors' Response**
>
> We thank the reviewer for their thoughtful comments.
>
> 1. The reviewer is absolutely correct; this analysis focuses on the adaptive/joint optimization aspect of adaptive dropout algorithms, rather than the dropout aspect. As such, we do not provide any new insight into the effects of dropout noise itself in general. However even standard non-adaptive dropout trivially falls into our framework by taking $p \to 2$ for the penalty $\frac{1}{p} \|\mathbf{w}\|_p^p$. As we show in Table 3 in the Appendix, this corresponds to a non-adaptive dropout strategy in the sense that $\hat{\eta}(w)$ loses its dependency on $w$.
>
> 2. The difficulty in going to losses other than the squared loss is that we typically cannot have exact expressions for the dropout regularizer. Even if we use expressions like those found in references [35] and [37] that we cite, they are only based on Taylor series approximations of the loss function, so there would be a degree of impreciseness in any such result, even for linear classification. The further difficulty is that even in these approximate dropout regularizers, they tend to deviate slightly from being purely quadratic in the weights, making it difficult to apply something like the $\eta$-trick. However, insofar as the implicit regularization due to dropout approximates diagonal ridge-like regularization, we believe that our analysis explains adaptive dropout in the classification setting as well. In addition, our experiment on variational dropout supports the idea that our theory also explains sparsification in the classification setting.
>
> 3. Regarding the question of whether the sparsity patterns are the same for each layer across the methods, yes, the sparsity patterns appear to be quite similar. Here is a table of the fraction of nonzeros for each layer at the end of training, which we will add to the appendix of the paper:
>
> | Net           | 784x300 | 300x100 | 100x10 |
> | ------------- | ------- | ------- | ------ |
> | VarDrop+LR+AR | 2.0     | 3.5     | 50.2   |
> | VarDrop+LR    | 7.2     | 18.9    | 99.9   |
> | VarDrop       | 56.8    | 78.8    | 100.0  |
> | $\eta$-Trick  | 5.4     | 2.6     | 20.6   |
> | AdaProx       | 2.6     | 2.4     | 39.9   |
> | AdaTikhonov   | 1.6     | 2.5     | 46.0   |
> | LogSum        | 1.6     | 2.5     | 47.9   |

---

### Official Review · Reviewer_QmiQ · 2021-07-21

**Rating:** 8
**Confidence:** 3

**Summary:**

This paper examines stochastic perturbation methods, such as dropout, through the lens of the "eta-trick".  This trick allows the optimization objective to be re-written as a joint objective in terms of the original parameters and auxiliary parameters eta.  This new objective consists of the original loss in addition to a quadratic penalty in the parameters and a formulation-specific function of eta.  The paper shows that any adaptive dropout method that updates the dropout parameters monotonically as a function of the weight magnitudes is equivalent to regularized risk minimization with a subquadratic penalty, which is known to induce sparsity.  The paper analyses standout, variational dropout, L0 regularization, and magnitude pruning from this perspective, deriving the corresponding penalty term (Table 2) and allowing for comparison.  The paper then experimentally compares variants of variational dropout, finding that performance is well-described by the eta-trick reformulation since stochastic variants suffer from the introduced noise.

**Limitations And Societal Impact:**

Yes, I found limitations and societal impact to have been addressed.

**Main Review:**

I found this paper well-written, interesting, and novel.  I had not seen the eta-trick before, and I found its use to compare seemingly disparate regularization methods to be fascinating.  Moreover, I like how the experiment isolates the different variants of variational dropout and compares them to the eta-trick-derived formulation.  I have no criticisms at this time; I find the paper ready for publication.

**Time Spent Reviewing:**

3

---

> ### Author Response · Authors · 2021-08-09
> **Many Thanks!**
>
> We thank the reviewer for very well summarizing our paper and contributions. Many thanks for appreciating our work!

---

### Official Review · Reviewer_t3cP · 2021-08-03

**Rating:** 6
**Confidence:** 4

**Summary:**

For linear regression, the paper shows that several popular heuristics used in deep learning correspond to regularized empirical risk minimization with sub-quadratic penalties, thereby, arguing that these methods indeed induce sparsity. The analysis leverages the so-called \eta-trick, where the penalty is replaced with its variational (minimization) form. Empirical results are also provided to support the theory.


**Limitations And Societal Impact:**

Yes

**Main Review:**

One of the strengths of the paper is that it unifies several popular heuristics, including variational dropout, standout, magnitude pruning, etc. The claims are rigorously proved and the empirical evidence validates some of the claims.

The main limitation of the paper is that \eta-trick itself is limited: it can be applied only when a weighted-sum-of-squared of parameters is present in the objective. Dropout in linear regression with l2-loss is known to induce such a penalty, which is leveraged here to make connections to several adaptive dropout scenarios. However, I'm not sure if the techniques can be generalized to analyze even simple 2-layer linear networks. The equivalent form of equation (4) has been studied for deeper linear models (2-layer and beyond). Given the more complicated form of the penalty in such models, do the authors hope this approach to generalize? Another limitation is that the paper merely focuses on the duality in the optimization domain. There is no statistical or computational analysis comparing either of the two forms (adaptive dropout or the dual IRLS). Is there a benefit in doing one over the other?

Although I’ve read several papers on the topic, following the paper was difficult at times. I think the current presentation makes it challenging for the average reader to understand the main results and constructions, and distinguish them from less important, tangential results. I suggest that the authors list the main contributions of the paper in the introduction. The duality between different penalties and their \eta-trick-representations in table 1 is interesting, but the description of their connection in sections 2.2 and 3 can be improved, in my opinion. The motivation behind studying adaptive dropout updates in section 3 is not clear at all. What is the merit of updates in equations (8) and (9), except for “facilitating your analysis” (as stated in line 123)?  Also, it is not clear how the “standardization” assumption affects the results and techniques. The authors suggest that “it is inexpensive to satisfy in practice, but our analysis can be extended to the general case”. However, they do not say if/how the main takeaways from the paper will be affected if the data is non-standard.

typo: line 20: trad-eoff

**Time Spent Reviewing:**

12

---

> ### Author Response · Authors · 2021-08-09
> **Authors' Response**
>
> We thank the reviewer for their thoughtful comments.
>
> 1. We understand that the reviewer is concerned about the general usefulness of the $\eta$-trick to understand regularization beyond the linear regression setting. We agree with the reviewer that there are very clear difficulties in directly applying this analysis beyond linear regression. However, we emphasize that our work enables the analysis of methods like variational dropout and “L0-norm regularization” for the first time, even in the linear case. These methods have had broad influence and hundreds of citations, yet neither of these works can be analyzed even in the linear regression case without the $\eta$-trick---that is, before our work.
>
>    That said, as we argue in the discussion section of the paper, we are hopeful that this connection offers a way to understand the regularizing effect of adaptive dropout strategies, if not as precisely as in linear regression. Please see our response to Reviewer 8X6v.
>
> 2. Regarding the question of statistical analysis and computational analysis, these would require the consideration of a particular algorithm (such as equation (9)), which is beyond the scope of our work. In the convex setting, aside from error due to algorithmic noise, statistical performance is determined solely by the optimization problem, which is covered by our analysis. (In the non-convex setting this is not necessarily true, but the statistical performance of algorithms solving non-convex problems of course remains an open problem in general.)
>
> 3. Regarding computational performance, we believe it is premature to make strong claims without significant empirical evidence, but our (unsubmitted) experiments actually suggest that at their best, adaptive dropout algorithms converge no faster than pure $\eta$-trick-type algorithms, and that at their worst they can exhibit slower convergence due to excessive noise. We discuss this briefly in Section 5 of the paper. We see hints of this in Figure 2, where vanilla variational dropout converges significantly more slowly than all of the other methods, which either reduce the variance due to dropout or eliminate it entirely.
>
> 4. Regarding standardized data, observe that only the diagonal elements of $\frac{1}{n}{\mathbf X}^\top{\mathbf X}$ affect the implicit penalty. A possible generalization for separable penalties is to introduce a vector of regularization parameters $(\lambda_1, \ldots, \lambda_p)$ such that $\lambda_i = \lambda / \sigma_i^2$, where $\sigma_i^2$ is the $i$th diagonal element of $\frac{1}{n}{\mathbf X}^\top{\mathbf X}$. Then each weight is regularized by the same regularizer $\Omega$, but each with its corresponding strength $\lambda_i$.

---

> ### Comment · Area_Chair_YN35 · 2021-09-04
> **Have the authors addressed your concerns?**
>
> Hi (t3cP),
>
> Can you please share your thoughts on authors response to your comments?
>
> Do you still think it is a borderline reject?
>
> Thanks!

---

> > ### Comment · Reviewer_t3cP · 2021-09-05
> >
> > Hi,
> > The authors have partially addressed my concerns, I've increased my score.
> > Thanks!

---

### Decision · Program_Chairs · 2021-09-27

**Decision:**

Accept (Poster)

**Comment:**

The paper shows that popular heuristics in deep learning, e.g. dropout, in the context of linear regression amount to regularizing the empirical objective with sub-quadratic penalties which promote sparsity. The paper introduces interesting tools that may be interesting outside the scope of the paper. Overall, a good paper.